# Urinary 8-OHdG as a Biomarker for Oxidative Stress: A Systematic Literature Review and Meta-Analysis

**DOI:** 10.3390/ijms21113743

**Published:** 2020-05-26

**Authors:** Melanie Graille, Pascal Wild, Jean-Jacques Sauvain, Maud Hemmendinger, Irina Guseva Canu, Nancy B. Hopf

**Affiliations:** 1Center for Primary Care and Public Health (Unisanté), University of Lausanne, Route de la Corniche, 21066 Epalinges-Lausanne, Switzerland; MelanieGraille@hotmail.com (M.G.); Pascal.Wild@inrs.fr (P.W.); jean-jacques.sauvain@unisante.ch (J.-J.S.); maud.hemmendinger@unisante.ch (M.H.); irina.guseva-canu@unisante.ch (I.G.C.); 2Institut national de recherche et de sécurité (INRS), 54000 Nancy, France; 3Swiss Center for Applied Human Toxicology (SCAHT), Missionsstrasse 64, 4055 Basel, Switzerland

**Keywords:** oxidative stress, biomarker, 8-OHdG, systematic review, meta-analysis

## Abstract

Oxidative stress reflects a disturbance in the balance between the production and accumulation of reactive oxygen species (ROS). ROS are scavenged by the antioxidant system, but when in excess concentration, they can oxidize proteins, lipids, and DNA. DNA damage is usually repaired, and the oxidized products are excreted in urine. 8-hydroxy-2-deoxyguanosine is considered a biomarker for oxidative damage of DNA. It is needed to define background ranges for 8-OHdG, to use it as a measure of oxidative stress overproduction. We established a standardized protocol for a systematic review and meta-analysis to assess background ranges for urinary 8-OHdG concentrations in healthy populations. We computed geometric mean (GM) and geometric standard deviations (GSD) as the basis for the meta-analysis. We retrieved an initial 1246 articles, included 84 articles, and identified 128 study subgroups. We stratified the subgroups by body mass index, gender, and smoking status reported. The pooled GM value for urinary 8-OHdG concentrations in healthy adults with a mean body mass index (BMI) ≤ 25 measured using chemical methods was 3.9 ng/mg creatinine (interquartile range (IQR): 3 to 5.5 ng/mg creatinine). A significant positive association was observed between smoking and urinary 8-OHdG concentrations when measured by chemical analysis. No gender effect was observed.

## 1. Introduction

Oxidative stress reflects a disturbance in the balance between the production and accumulation of reactive oxygen species (ROS), and an overproduction of ROS has negative consequences for cell physiology [1]. When ROS concentration is in excess, oxidative damage to proteins, lipids, and DNA occurs, thus causing structural and functional cellular changes. DNA damage is usually repaired primarily via the base excision repair pathway, and oxidized products are excreted in urine [2]. 8-hydroxy-2-deoxyguanosine (8-OHdG) is one of the most widely studied oxidized metabolites and is considered as a biomarker for oxidative damage of DNA [3,4]. The formation of 8-OHdG by oxygen radicals was first reported in 1984 by Kasai and Nishimura [5].

The interaction of the hydroxyl radical, the most important oxygen-free radical, with the nucleobases of the DNA strand, such as guanine, leads to the formation of 8-OHdG [6] (Figure 1).

Some diseases, such as cardiovascular or chronic obstructive pulmonary diseases (COPD), have been associated with excessive concentrations of 8-OHdG [7,8]. 8-OHdG levels also increase due to smoking, aging, or occupational exposure to physical, chemical, or biological substances [9,10].

A recent study suggested that 8-OHdG had high intraclass correlation coefficients (0.96), reproducible measurements, and low coefficients of variation and was the most suitable biomarker of oxidative stress in spot urine samples [11]. Concentrations of urinary oxidative stress biomarkers have been proposed as an effect biomarker to survey populations exposed to xenobiotics such as particulates, oxidizing agents, and lately, engineered nanomaterials [12,13].

Measuring urinary 8-OHdG has some advantages as it is very stable in urine [14], it is noninvasive, and its excretion is likely to reflect the oxidative DNA damage [15] and can be assessed by two main analytical techniques: mass-based methods (using either gas (GC) or liquid (LC) chromatography) and immunological methods. Another source of 8-OHdG in urine is DNA polymerase-dependent incorporation of 8-oxodGTP from the nucleotide pool [16]. Chromatographic methods are considered to be the gold standard; however, immunological techniques, which are less costly and time-consuming, are widely used because enzyme-linked immunosorbent assay (ELISA) kits have been developed for rapid detection and quantification of 8-OHdG [14,17].

A background range for 8-OHdG has been reported in different studies for healthy persons [11,18,19,20]. However, these studies reported a wider range of values, making the identification of background cut-off values challenging.

Therefore, the systematic review and meta-analyses of the reported values appears the most appropriate approach to bypass this issue. Our objective was to assess background ranges for urinary 8-OHdG concentrations in healthy adults.

## 2. Results

Chemical methods were used in 44 of the 128 study subgroups, and immunological techniques were used in 84 (Table 1). We decided to stratify the subgroups by body mass index (BMI), gender, and smoking status reported.

### 2.1. Descriptive Results

We retrieved 1246 articles, included 84 articles, and considered 129 study subgroups (Figure 2, Table 2, Table 3, Table 4 and Table 5) in the quantitative synthesis, which we stratified by main quantification techniques: immunological and chemical methods. For subgroups evaluated with the chemical methods, 31 studies had participants with a mean BMI between 18 and 25 (14 study subgroups of nonsmokers and 2 study subgroups of smokers) (Figure 3, Table 2). Nine studies had participants with a mean BMI > 25 (three study subgroups of nonsmokers and two study subgroups of smokers) (Figure 4, Table 3). The mean BMI was unknown for four study subgroups.

For subgroups analyzed with immunological techniques, 47 studies had participants with a mean BMI between 18 and 25 (24 study subgroups of nonsmokers, no study subgroups of smokers and 6 study subgroups with unknown smoking status) (Figure 5, Table 4). Twenty-six studies had participants with a mean BMI > 25 (13 study subgroups of nonsmokers and 6 study subgroups of smokers) (Figure 6, Table 5). The mean BMI was unknown for 11 study subgroups. Appendix A provides detailed information on the criteria used for the quality assessment (S1) and on the quality level of each included study subgroup (S2). Overall, two study subgroups (1.8%) were classified as low quality, 66 (58.4%) as moderate quality, and 45 study subgroups (39.8%) were of high quality.

### 2.2. Meta-Analysis Results 

As between-study heterogeneity was much larger than the between-subject heterogeneity, we decided to use a mixed model with study ID as a random effect. The IQR of subgroup-specific GM in subgroups with a mean BMI ≤ 25 with 8-OHdG measured using chemical methods was 3 to 5.5 ng/mg creatinine (Table 1). IQR of subgroup-specific GM in subgroups with a mean BMI > 25 measured using immunological methods was 5.9 to 19.8 ng/mg creatinine (Table 1).

We compared urinary 8-OHdG concentrations by smoking status within the study subgroups analyzed with chemical techniques and found that for study subgroups with mean BMI ≤ 25, smokers were 2.84 ([2.56, 3.16], *p* < 0.0001) times greater compared to nonsmoker study subgroups.

For study subgroups with mean BMI > 25, smokers were 1.61 ([1.17, 2.23], *p* = 0.004) times greater compared to the nonsmoker study.

No consistent effects of BMI and gender were observed in our mixed model either for chemical or immunological methods. Gender and BMI seem to not influence urinary 8-OHdG concentrations.

## 3. Discussion

### 3.1. Interpretation of Findings

We found that urinary 8-OHdG concentrations in smokers were greater than in nonsmokers when analysis was conducted with chemical techniques. However, in the population with mean BMI between 18 and 25, this finding was mainly due to one study [32] and needs to be confirmed. The absence of BMI effect on 8-OHdG in urine is in line with data from Lee et al. 2010 [93].

The IQR range for 8-OHdG in urine given in this meta-analysis is in line with two other studies trying to define reference values for the Italian population (female: 3.25–6.85 ng/mg creatinine; male: 2.9–5.5 ng/mg creatinine) [94]. The absence of gender effect observed for 8-OHdG in this study is in line with data from the Italian population [94] but in contradiction with two others [93,95].

The analysis of the data was difficult due to the diversity in study design, analytical methods (chemical or immunoassay techniques), statistical analysis, and data presentation in studies included.

### 3.2. Quantification of 8-OHdG

The heterogeneity in techniques used to quantify urinary 8-OHdG makes it more difficult to compare data between laboratories.

Chemical techniques are superior to immunological techniques due to their sensitivity and specificity [14,96]. Chemical techniques require expensive instruments and trained users, but we recommend using chemical quantification methods as standard methods for future studies of biomonitoring.

### 3.3. Lack of Homogeneity in Data Collection and Reporting

Most studies used spot urine samples for 8-OHdG rather than 12- or 24-h collection. However, 8-OHdG levels showed fluctuation during the day under oxidative states [97], but good correlations have been observed between levels of 8-OHdG in spot morning urine and levels of 8-OHdG in the 24-h urinary collection [14]. Therefore, we included studies reporting spot morning urine, 12- or 24-h urinary samples. The first morning urine void is particularly valuable because it provides a time average for biomarker concentrations that may occur during the hours of sleep (approximately 8 h) and is also relatively free of dietary, physical, and environmental exposures [15]. A significant increase in time in the urinary 8-OHdG during the first part of the day was recently reported among smokers [15]. To make it easier to compare results between studies, we recommend collecting spot morning urine.

### 3.4. Limitations

We confirm that smokers have a significantly greater concentration of urinary 8-OHdG, as has been previously reported in the literature. The concentration differences need to be quantified, but with only a few studies in smokers available, this cannot be done at the present time.

We emphasize here that the values we report are for a healthy population. We were not able to analyze parameters previously reported to influence 8-OHdG concentrations such as occupation, pregnancy, special diet, vitamin, and physical activity due to the limited number of studies with such data.

### 3.5. Recommendations

The fluctuation in urine flow rate could in fact affect the assessment of urinary 8-OHdG. The urinary 8-OHdG concentrations need to be normalized by urinary creatinine concentrations for healthy adults. Different studies indicated a correlation between excretion of creatinine and 8-OHdG [94,95]. In addition, normalization with creatinine for spot urine can be considered as a surrogate for the 24-h excretion of 8-OHdG [94,98].

To reach consensual background of urinary 8-OHdG values, harmonization of the unit (ng/mg creatinine) is needed. Harmonization of the statistical reporting of the results is also recommended (geometric means (GM) and geometric standard deviations (GSD)). We suggest reporting the median and the 1st and 3rd quartile as GSDs are not easy to interpret.

## 4. Materials and Methods

We established a standardized protocol for systematic review and meta-analysis for a set of biomarkers of oxidative stress. This protocol was registered in the International Prospective Register of Systematic Reviews (registration number CRD 42020146623) [99] and described in detail by Hemmendinger et al. [100]. The protocol was then adapted for each biomarker depending on the biological matrix focused, here the urinary 8-OHdG. The methods and results of this study are reported following recommendations from Preferred Reporting Items for Systematic Reviews and Meta-Analyses (PRISMA) recommendations [101,102].

### 4.1. Literature Search

The search strategy was done with a medical librarian. The MeSH (Medical Subject Headings) terms from the PubMed database and free text words were combined. The complete search string was: (“Smoking/urine”[Mesh] OR “Urine”[Mesh] OR Urine*[tiab] OR Urinary[tiab] OR Urinal*[tiab]) AND (“8-oxo-7-hydrodeoxyguanosine”[Supplementary Concept] OR 8-OHdg[tw] OR 8ohdg[tw] OR 8-oh-dg[tw] OR 8-ohg[tw] OR 8-OH-2dG[tw] OR 8-hydroxydeoxyguanosine[tw] OR 8-hydroxyguanine[tw] OR 8-hydroxy-g[tw] OR 8-hydroxy-dg[tw] OR 8-hydroxy-guanine[tw] OR 8-hydroxy-2-deoxyguanosine[tw]) NOT ((“Child”[Mesh] OR “Infant”[Mesh] OR “Adolescent”[MeSH]) NOT “adult”[MeSH]) NOT (animals[mh] NOT humans[mh]).

### 4.2. Study Selection

The search was performed on 7 May 2019. Rayyan [103], a systematic review web application, was used for title and abstract screening. We selected the studies in a stepwise process as depicted in Figure 6. To be included in the analysis, a study had to be in English and to provide urinary 8-OHdG concentrations in healthy adults (ages 18—no upper age limit) populations. We excluded non-human studies, in vitro studies, reviews, letters, expert opinions, and editorials. We read the eligible articles in depth, and only studies with original data from healthy (no known disease) adult populations were included in the statistical analysis. All techniques used for the quantification of 8-OHdG were included and classified accordingly. We excluded studies with coefficient variation <10% or >200%. We also excluded data suspected to have unit or reported value mistakes (more than three orders of magnitude higher than the median levels).

### 4.3. Data Extraction

We extracted the following information: first author name, publication year, study type, country, analytic method, sample time, sample size, gender, mean age, mean BMI, smoking status, season, occupation, pregnancy, diet, vitamin, exercise, outcome (8-OHdG concentration), references, and article DOI. We extracted all subgroup-specific data when data on several subgroups were available in a given paper. Then, we excluded all subgroups selected based on disease status (e.g., cardiovascular disease) and all subgroups selected based on an exposure status (e.g., bus drivers). If data on the same subgroup were reported for different times (e.g., different seasons), only the data at the time of participant inclusion were included. In a third round, we excluded duplicate data (e.g., control population reported in more than one study) and retained the most complete and the most recent study.

### 4.4. Statistical Analysis

First, we analyzed the values of urinary 8-OHdG measured in original studies in view of establishing the background ranges using meta-analysis. Measured values were generally log-normally distributed. We therefore computed geometric means (GM) and geometric standard deviations (GSD) as the basis for the meta-analysis or equivalently muL=ln(GM) and sdL=ln(GSD). Further details on the data treatment and analyses are available elsewhere [104].

We could not compute standard errors on the geometric (or arithmetic) scale when neither standard deviation (SD), GSD, IQR, nor confidence interval (CI) were reported. As a consequence, we excluded these studies from the meta-analysis. We converted all the concentration values to the same units (ng/mg creatinine) before computing GM and GSD. We used 113.12 g/mol for the molecular weight for creatinine and 283.24 g/mol for 8-OHdG. We regrouped the data according to analytical techniques used; immunological techniques and chemical techniques. The data were analyzed separately.

We followed standard practice in meta-analysis [105] and represent the data as forest plots including the I-squared. This is an estimate of the between-study heterogeneity in percentage. If the between-study heterogeneity is much larger than the between-subject heterogeneity, then I^2^ is large. In this case, any attempt of obtaining a background value for individual participants will not be valid. In our case, a mixed model with study ID as a random effect is a more relevant analysis model. This yields results on the study subgroup level rather than at the individual level. Data management and statistical analyses were performed in STATA version 16 software.

## 5. Conclusions

We report pooled GM values for urinary 8-OHdG in healthy adults, separately for chemical and immunological methods. We observed a significant positive association between smoking status and urinary 8-OHdG concentrations when measured by chemical analysis. No gender effect was shown. Urinary 8-OHdG can potentially be used to quantify excess oxidative stress due to external exposures when background values have been established in different populations. We recommend adjusting urine samples with creatinine, quantifying 8-OHdG with chemical methods, and reporting results as median and quartiles. Comparing values across studies will then be feasible.

## Figures and Tables

**Figure 1 ijms-21-03743-f001:**
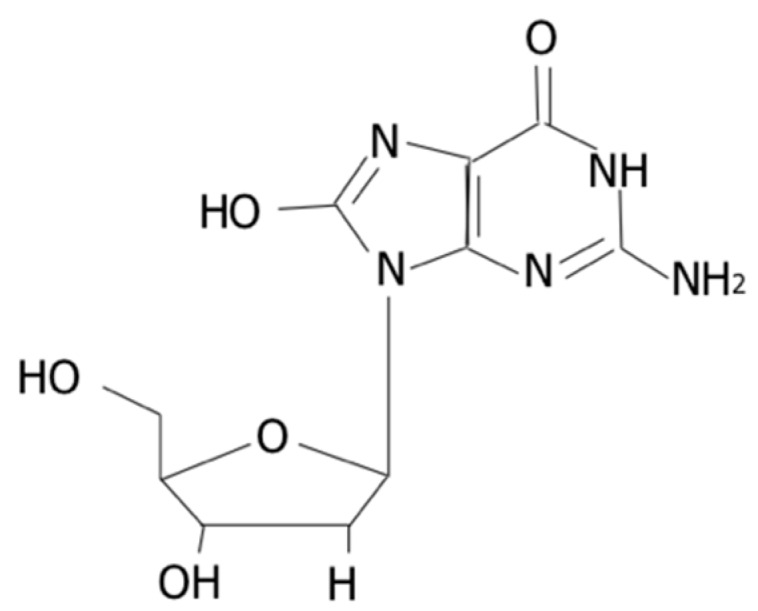
Structure of 8-OHdG.

**Figure 2 ijms-21-03743-f002:**
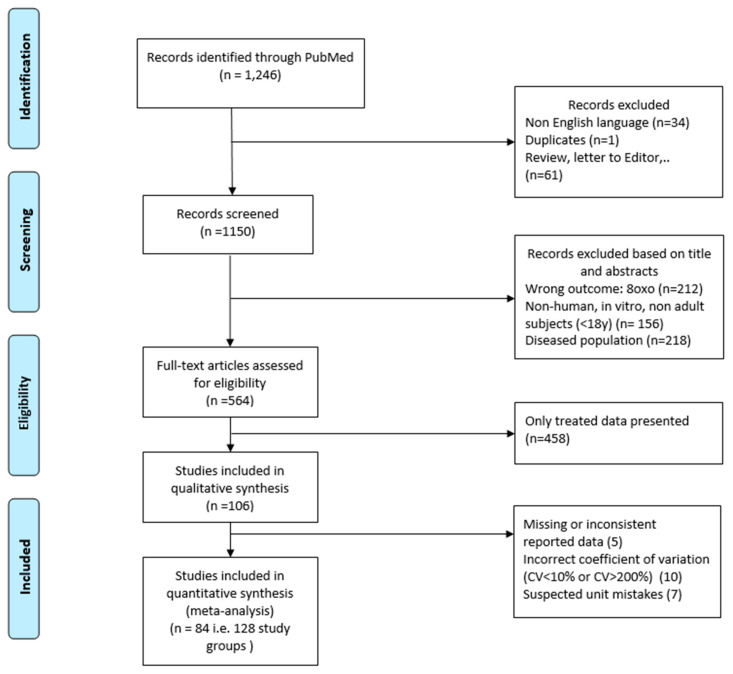
Flow chart of study selection.

**Figure 3 ijms-21-03743-f003:**
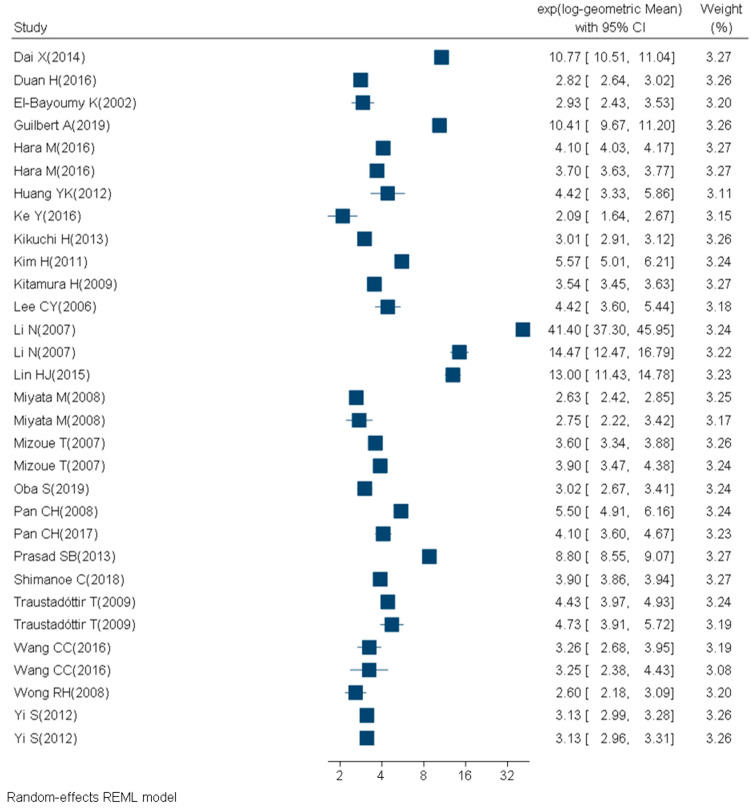
Forest plot of urinary 8-OHdG concentrations (ng/mg creatinine) measured with chemical techniques in healthy (mean BMI ≤ 25 and no known disease), adult (18+ years) participants.

**Figure 4 ijms-21-03743-f004:**
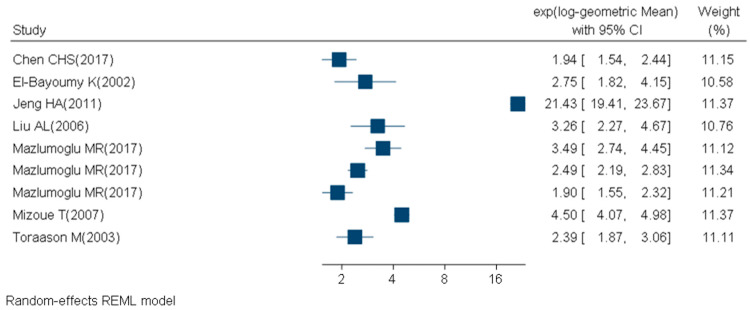
Forest plot of urinary 8-OHdG levels (ng/mg creatinine) measured with chemical techniques in healthy (mean BMI > 25 and no known disease), adult (18+ years) participants

**Figure 5 ijms-21-03743-f005:**
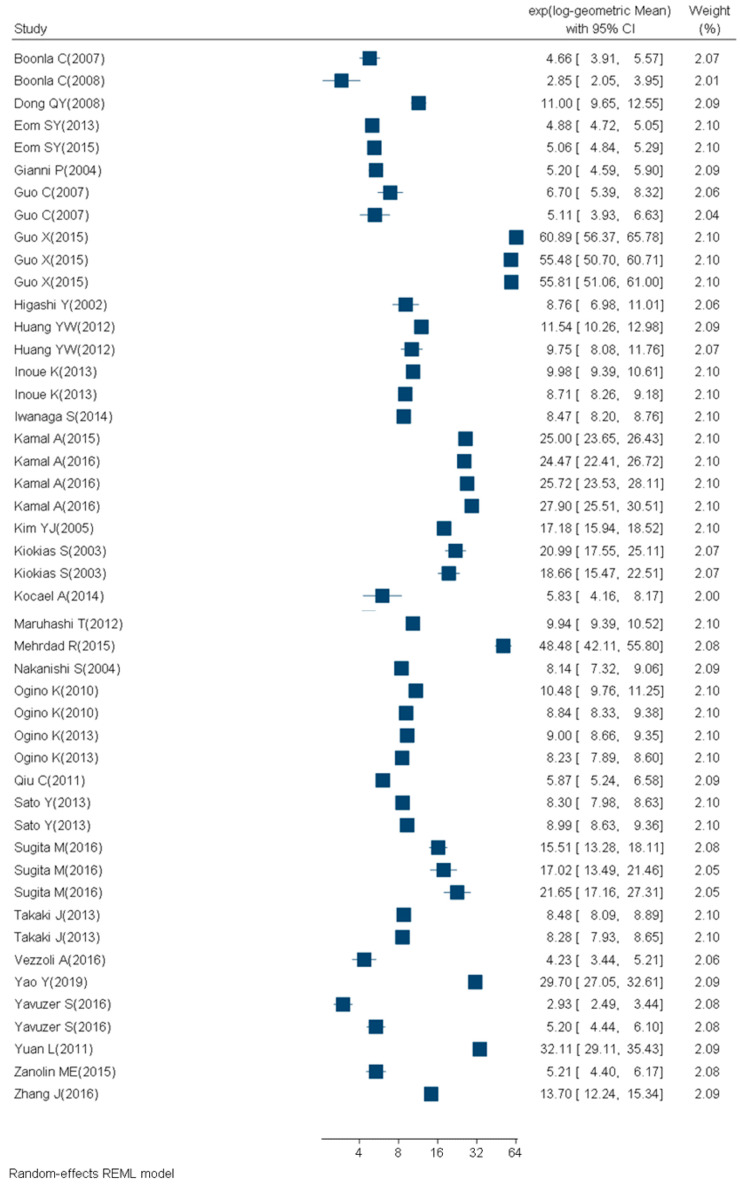
Forest plot of urinary 8-OHdG concentrations (ng/mg creatinine) measured with immunological techniques in healthy (mean BMI ≤ 25 and no known disease), adult (18+ years) participants.

**Figure 6 ijms-21-03743-f006:**
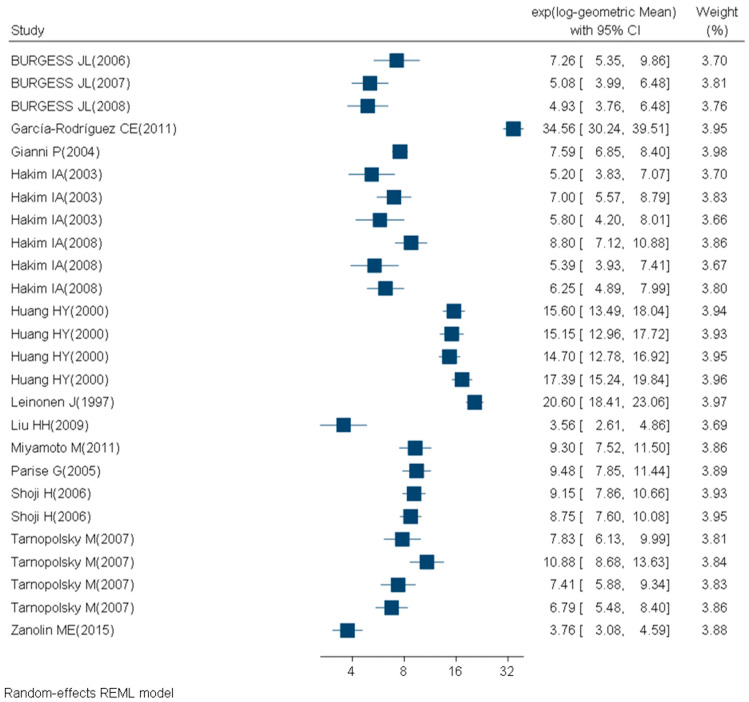
Forest plot of urinary 8-OHdG concentrations (ng/mg creatinine) measured with immunological techniques in healthy (mean BMI > 25 and no known disease), adult (18+ years) participants.

**Table 1 ijms-21-03743-t001:** Summary of geometric mean urinary 8-OHdG concentrations (ng/mg creatinine) in subgroups of healthy adult (18+ years) participants.

	BMI ≤ 25			BMI > 25		
Analytical Techniques	All Participants	Smoking Status		All Participants	Smoking Status	
Chemical	3.9 * (3–5.5) (*n* ** = 31)	Nonsmokers	4.3	2.8 (2.4–3.5) (*n* = 9)	Nonsmokers	2.5
		(*n* = 14)	(2.9–5.5)		(*n* = 3)	(1.9–2.8)
		Smokers	22.2		Smokers	4.0
		(*n* = 2)	(3–41.4)		(*n* = 2)	(3.5–4.5)
Immunological	9.0 (5.9–19.8) (*n* = 47)	Nonsmokers	11.5	7.7 (5.8 – 10.9) (*n* = 26)	Nonsmokers	9.3
		(*n* = 24)	(5.9–21.6)		(*n* = 13)	(7.8–14.7)
		Smokers	NA		Smokers	6.0
		(*n* = 0)			(*n* = 6)	(5.4–7)

* Median (IQR: 25%–75%); ** Number of included study subgroups; NA: Not Available.

**Table 2 ijms-21-03743-t002:** References for urinary 8-OHdG concentrations and computed GM (ng/mg creatinine) measured with chemical techniques in healthy (mean BMI ≤ 25 and no known disease), adult (18+ years) participants.

Reference	Study Group	Analytic Method	Sample	Country	Sample Size	Mean Age	Female	Male	Smoking Status	Mean BMI	AM	GM	IQR	Range	Median	CI	Units (8-OHdG/Creatinine)	Computed GM (ng/mg Creatinine)	GSD
[21]	Control group	HPLC	spot urine	China	497	42.48	113	384	50%	23.72	4.47 ± 1.26 *						nmol/mmol	11	1
[22]	Control group	HPLC-MS/MS	spot urine	China	106	31.62	0	106	52.8%	23.79	3 ± 1.08 *						μg/g	3	1
[23]	Selenium group baseline value	HPLC with EC detection	spot urine	USA	17	30.7	0	17	0%	24.2	3.16 ± 1.28 *						ng/mg	3	1
[24]	Baseline value	UPLC-MS-MS in positive EI mode	spot urine	Belgium	48	40	3	45	31.2%	24.2	10.76 ± 2.83 *			7.05–20.92			μg/g	10	1
[25]	Men baseline	HPLC	spot urine	Japan	2370	60.7	0	2370	24.9%	23.6		3.7 ± 1.6 *					ng/mg	4	2
[25]	Women baseline	HPLC	spot urine	Japan	4052	60.2	4052	0	4.7%	22.2		4.1 ± 1.7 *					ng/mg	4	2
[26]	Baseline value	LC-MS/MS	spot urine	Taiwan	58	23.84	0	58	51.7%	24.55			2.63–11.54		4.42		µg/g	16	2
[27]	Service staff group	HPLC	spot urine	China	67	24.8	0	67	0%	23.2					1.4	0.9–1.8	μmol/mol	2	3
[28]	All population	HPLC	spot urine	Japan	503	42.4	209	294	27.4%	22.5			2.37–4.03	0.8–10.0	3.01		μg/g	3	1
[29]	Baseline value	HPLC	spot urine	Korea	102	55	102	0	0%	24.1	6.5 ± 3.9 *						μg/g	6	2
[30]	Control group	HPLC	spot urine	Japan	805	40.3	0	805	46.7%	23.7	3.79 ± 1.44 *						ng/mg	4	1
[31]	Baseline value	GC-MS	spot urine	Singapore	24	22.8	NA	NA	0%	21.6	2.02 ± 1.12 *						µmol/mol	5	2
[32]	Non smoker group baseline	HPLC with EC detection	24 h urine	China	30	21.5	0	30	0%	22.8	6.3 ± 0.5 **						µmol/mol	14	2
[32]	Smoker group baseline	HPLC with EC detection	24 h urine	China	60	21.8	0	60	100%	22.6	18 ± 1 **						µmol/mol	41	2
[33]	Participants without stroke	LC–MS/MS	spot urine	Taiwan	131	64.9	57	74	50%	22.9			8.3–22.8		13		μg/g	13	2
[34]	Male baseline	HPLC-EC	spot urine	Japan	79	47.9	0	79	0%	22.3	2.81 ± 1.07 *						μg/g	3	2
[34]	Female baseline	HPLC-EC	spot urine	Japan	16	46.7	16	0	0%	20.6	3.04 ± 1.42 *						μg/g	3	1
[35]	Women baseline	HPLC	spot urine	Japan	37	28–57	37	0	5.4%	21.5			3.2–5.2		3.9		μg/g	4	1
[35]	Non smoking men group baseline	HPLC	spot urine	Japan	87	28–57	0	87	0%	24.2			2.9–4.7		3.6		μg/g	4	1
[36]	Baseline value	HPLC with an EC detector	spot urine	Japan	23	46.8	11	12	100%	23.6		3.02				2.24–4.07	ng/mg	5	1
[37]	Control group	LC–MS/MS	spot urine	Taiwan	125	34.1	0	125	0%	22.8		4.1 ± 2.1 *					μg/g	4	2
[38]	Control group	HPLC-MS/MS	spot urine	China	185	40.4	124	61	0%	24.4		5.5 ± 2.2 *					µg/g	6	2
[39]	Control group	LC EC	spot urine	India	135	41.31	0	135	0%	22.38	3.57 ± 0.63 *						μmol/mol	9	1
[40]	All population	HPLC	spot urine	Japan	6517	60.3	4064	2453	12.6%	22.7		3.9 ± 1.6 *					ng/mg	4	2
[41]	Placebo group end of study	LC-MS/MS	spot urine	USA	12	69	6	6	0%	25	2 ± 0.2 **						µmol/mol	5	1
[41]	Tart cherryjuice group end of study	LC-MS/MS	spot urine	USA	12	69	6	6	0%	25	1.8 ± 0.1 **						µmol/mol	4	1
[42]	Male group	LC-MS/MS	spot urine	China	69	37.83	0	69	43.5%	24.1	4.55 ± 4.44 *						μg/g	3	2
[42]	Female group	LC-MS/MS	spot urine	China	23	38.55	23	0	0%	22.1	4.34 ± 3.85 *						μg/g	3	2
[43]	Control group	LC–MS/MS	spot urine	Taiwan	129	51.7	39	90	27.9%	24.6	4.3 ± 0.5 **						ng/mg	3	3
[44]	Men group	HPLC	spot urine	Japan	196	44.4	0	196	43.9%	23.8	3.3 ± 1.1 *						µg/g	3	1
[44]	Women group	HPLC	spot urine	Japan	136	40.4	136	0	2.9%	21	3.3 ± 1.1 *						µg/g	3	1

* SD, ** SEM.

**Table 3 ijms-21-03743-t003:** References for urinary 8-OHdG concentrations measured and computed GM (ng/mg creatinine) with chemical techniques in healthy (mean BMI > 25 and no known disease), adult (18+ years) participants.

Reference	Study Group	Analytic Method	Sample	Country	Sample Size	Mean Age	Female	Male	Smoking Status	Mean BMI	AM	IQR	Median	Units (8-OHdG/Creatinine)	Computed GM (ng/mg Creatinine)	GSD
[45]	Elderly low expose group	LC–MS/MS	spot urine	Taiwan	71	66.36	36	35	9.9%	26.36	3.16 ± 4.07 *			μg/g	3	2
[23]	Placebo group	HPLC with EC detection	spot urine	USA	19	31.1	0	19	0%	25.2	4.18 ± 4.78 *			ng/mg	3	2
[46]	Control group	LC/MS/MS	spot urine	Taiwan	168	43.2	NA	NA	34%	26.4	10.61 ± 7.77 *			µmol/mol	21	2
[47]	Control group	HPLC	spot urine	China	31	38.7	0	31	19.4%	≤24 38.7% >24 61.3%		1.0–4.0	1.3	µmol/mol	3	3
[48]	Control non smoker group	HPLC–ECD	spot urine	Turkey	19	54.8	3	16	0%	29.1	2.1 ± 1 *			μg/g	2	1
[48]	Control ex-smoker group	HPLC–ECD	spot urine	Turkey	21	57.5	3	18	0%	27.2	2.6 ± 0.8 *			μg/g	2	2
[48]	Control smoker group	HPLC– ECD	spot urine	Turkey	24	51.1	4	20	100%	26.5	4.2 ± 2.8 *			μg/g	3	2
[35]	Smoking men group baseline	HPLC	spot urine	Japan	40	28–57	0	40	100%	25.1		3.6–5.6	4.5	μg/g	4	1
[49]	Control group baseline	HPLC	spot urine	USA	20	39	20	0	50%	29	2.8 ± 1.7 *			µg/g	2	2

* SD.

**Table 4 ijms-21-03743-t004:** References for urinary 8-OHdG concentrations measured and computed GM (ng/mg creatinine) with immunological techniques in healthy (mean BMI ≤ 25 and no known disease), adult (18+ years) participants.

Reference	Study Group	Sample	Country	Sample Size	Mean Age	Female	Male	Smoking Status	Mean BMI	AM	GM	IQR	Range	Median	CI	Units (8-OHdG/Creatinine)	Computed GM (ng/mg Creatinine)	GSD
[50]	Healthy control group	24 h urine	Thailand	30	41.43	19	11	NA	22.56	4.32 ± 4.93 *						μg/g	3	2
[51]	Healthy control group	24 h urine	Thailand	30	41.43	19	11	NA	22.56	5.27 ± 2.77 *						μg/g	5	2
[52]	Control group	spot urine	China	35	60	15	20	0%	22.9	11.9 ± 4.9 *						ng/mg	11	1
[53]	Control group	spot urine	Korea	416	64.4	92	324	28.1%	23.7		5.06				4.55–5.62	μg/g	5	2
[54]	Control group	spot urine	Korea	140	68.8	32	108	65.5%	22.46		4.88				4.43–5.38	μg/g	5	1
[55]	Healthy young group	24 h urine	Canada	12	22.8	0	12	0%	25	5333 ± 1191 *						ng/g	5	1
[56]	Apple group final value	spot urine	China	13	62.8	3	10	0%	24.2	824.41 ± 343.66 *						ng/mmol	7	1
[56]	Pomegranate group final value	spot urine	China	13	64.1	3	10	0%	23	651.57 ± 332.44 *						ng/mmol	5	2
[57]	placebo group baseline value	spot urine	China	150	51.58	92	58	41.3%	23.8		60.89 ± 1.62*			58.19		ng/mg	61	2
[57]	Baseline line value Low FA group	spot urine	China	145	48.9	87	58	33.8%	24.5		55.48 ± 1.74 *			53.51		ng/mg	55	2
[57]	Baseline value high FA group	spot urine	China	143	48.66	94	49	30.1%	24.6		55.81 ± 1.72 *			54.73		ng/mg	56	2
[58]	Control group	24 h urine	Japan	15	40	6	9	0%	23.2	9.7 ± 4.6 *						ng/mg	9	2
[59]	Control group I	spot urine	China	20	25.55	17	3	0%	19.74	10.68 ± 1.07 **						ng/mg	10	2
[59]	Control group II	spot urine	China	20	24.5	15	5	0%	20.09	11.96 ± 0.73 **						ng/mg	12	1
[60]	Male group	spot urine	Japan	195	41.7	0	195	49.7%	23.6	9.35 ± 3.66 *						ng/mg	9	1
[60]	Female group	spot urine	Japan	194	41.7	194	0	29.9%	22.1	10.97 ± 5 *						ng/mg	10	2
[61]	Non MS group	spot urine	Japan	638	40.8	385	253	27.3%	22.3	9.28 ± 4.15 *						ng/mg	8	2
[62]	Male control	spot urine	Pakistan	34	39.7	0	34	0%	19.85	24.5 ± 6.6 *			11.08–33.85	25.72		ng/mg	26	1
[62]	Female control	spot urine	Pakistan	32	39.52	32	0	0%	20.83	24.5 ± 6.33 *			11.1–33.85	24.47		ng/mg	24	1
[63]	Control group	spot urine	Pakistan	34	39.7	0	34	0%	20.9	24 ± 4 *			9–30	25		ng/mg	25	1
[64]	Control group	spot urine	Pakistan	34	37	0	34	0%	20.8	25.8 ± 7 *			9.1–33.9	27.9		ng/mg	28	1
[65]	Pregnant women	spot urine	Korea	261	29.6	261	0	0%	21	20.8 ± 14.2 *						µg/g	17	2
[66]	Control group baseline	spot urine	UK	32	31.7	15	17	0%	22.4	21.6 ± 12.6 *						ng/mg	19	2
[66]	Test group baseline	spot urine	UK	32	31.7	15	17	0%	22.4	24 ± 13.3 *						ng/mg	21	2
[67]	Control group	spot urine	Turkey	20	40.7	10	10	NA	22.52	7.84 ± 7.04 *						ng/mg	6	2
[68]	Control group	spot urine	Japan	108	23	0	108	NA	22.5	10.4 ± 3.2 *						ng/mg	10	1
[69]	Non exposed group	spot urine	Iran	43	35.58	0	43	21%	19–24	54.16 ± 26.98 *						ng/mg	48	2
[70]	Control group	spot urine	Japan	52	62.4	27	25	0%	24	8.8 ± 0.5 **						ng/mg	8	1
[71]	Male group	spot urine	Japan	276	42.1	0	276	NA	23.8	8.8 ± 0.2 **						ng/mg	8	1
[71]	Female group	spot urine	Japan	445	42.7	445	0	NA	21.9	9.8 ± 0.2 **						ng/mg	9	2
[72]	Male healthy population	spot urine	Japan	142	43.6	0	142	31%	22.4	11.5 ± 5.2 *						ng/mg	10	2
[72]	Female healthy population	spot urine	Japan	136	43.4	136	0	52.2%	23.8	9.4 ± 3.4 *						ng/mg	9	1
[73]	Control group	spot urine	USA	43	32.6	43	0	0%	23.2	6.31 ± 2.49 *						ng/mg	6	1
[74]	Male group	spot urine	Japan	323	42	0	323	42.7%	23.7	8.85 ± 3.29						ng/mg	8	1
[74]	Female group	spot urine	Japan	443	42.7	443	0	13.5%	21.9	9.89 ± 4.54 *						ng/mg	9	2
[75]	Green tea catechin-no exercise group baseline value	spot urine	Japan	8	22.4	0	8	0%	>18 <25	15.9 ± 3.6 *						ng/mg	16	1
[75]	Green tea catechin-exercise group baseline value	spot urine	Japan	8	21.1	0	8	0%	>18 <25	22.9 ± 7.9 *						ng/mg	22	1
[75]	Placebo group	spot urine	Japan	8	21.1	0	8	0%	>18 <25	18 ± 6.2 *						ng/mg	17	1
[76]	Men group	spot urine	Japan	272	43.5	0	272	60.7%	23.7	8.86 ± 3.36 *			2.13–21.87			μg/g	8	1
[76]	Women group	spot urine	Japan	295	40.3	295	0	15.6%	21.7	9.25 ± 4.03 *			0.05–25.56			μg/g	8	2
[77]	Baseline value 50km group	spot urine	Italy	6	41.83	NA	NA	0%	21.08	4.38 ± 1.16 *						ng/mg	4	1
[78]	Summer Non heating season	spot urine	China	34	47.9	34	0	0%	23.2	12.7 ± 4.7 *			2.60, 25.8	13.6		ng/mg	9	2
[79]	Healthy volunteers Young group	spot urine	Turkey	30	41.6	22	8	0%	22.1	3.24 ± 1.54 *						ng/mg	3	2
[79]	Healthy volunteers Elderly group	spot urine	Turkey	30	69.1	20	10	0%	23.6	5.74 ± 2.68 *						ng/mg	5	2
[80]	Baseline value	spot urine	China	25	20.9	12	13	0%	20.67	3765.63 ± 958.14 *						ng/mmol	32	1
[15]	Women group	spot urine	Italy	33	30	33	0	29%	20.7			3.68–7.20		5.21		ng/mg	4	2
[81]	Non exposed group	spot urine	China	143	27.89	100	43	8%	21.03	17.36 ± 13.5 *						ng/mg	14	2

* SD; ** SEM.

**Table 5 ijms-21-03743-t005:** References for urinary 8-OHdG concentrations measured and computed GM (ng/mg creatinine) with immunological techniques in healthy (mean BMI > 25 and no known disease), adult (18+ years) participants.

Reference	Study Group	Sample	Country	Sample Size	Mean Age	Female	Male	Smoking Status	MeanBMI	AM	IQR	Range	Median	Units (8-OHdG/Creatinine)	Computed GM (ng/mg Creatinine)	GSD
[82]	Cocorit communities	spot urine	Mexico	10	45.9	5	5	30%	27	8.2 ± 4.3 *				μg/g	7	2
[82]	Pueblo Yaqui communities	spot urine	Mexico	15	35.3	9	6	7%	26.7	5.7 ± 2.9 *				μg/g	5	2
[82]	Campo 47	spot urine	Mexico	15	39.5	10	5	40%	29.8	5.7 ± 3.3 *				μg/g	5	2
[83]	Control group	spot urine	UK	61	28.4	61	0	9%	26	39.83 ± 2.92 **				ng/mg	35	2
[55]	Healthy older group	24 h urine	Canada	12	71.8	0	12	0%	28.8	7714 ± 1402 *				ng/g	8	1
[84]	Water group baseline value	spot urine	USA	42	18–79	32	10	100%	25.9	8.7 ± 1.3 **				ng/mg	5	3
[84]	Green tea group baseline value	spot urine	USA	35	18–79	27	8	100%	26.5	10.8 ± 1.3 **				ng/mg	9	2
[84]	Black tea baseline value	spot urine	USA	43	18–79	31	12	100%	26.7	9.5 ± 2.1 **				ng/mg	6	2
[85]	Water group baseline value	spot urine	USA	45	49.8	32	13	100%	26.9	9.5 ± 1.3 **				ng/mg	6	3
[85]	Black tea baseline value	spot urine	USA	46	52.1	34	12	100%	27.2	10.8 ± 2.5 **				ng/mg	7	2
[85]	Green tea group baseline value	spot urine	USA	42	51.6	32	10	100%	27.2	8.7 ± 1.8 **				ng/mg	5	3
[86]	Placebo group baseline value	24 h urine	USA	47	58.1	23	24	0%	28.9	17.6 ± 10.4 *				ng/mg	15	2
[86]	Vit C group baseline value	24 h urine	USA	46	61.2	26	20	0%	28.7	19.3 ± 9.3 *				ng/mg	17	2
[86]	Vit E group baseline value	24 h urine	USA	45	55.5	29	16	0%	28.6	16.5 ± 8.4 *				ng/mg	15	2
[86]	Vit C + Vit E baseline value	24 h urine	USA	46	57.7	24	22	0%	28.9	17.7 ± 9.5 *				ng/mg	16	2
[87]	Control group	24 h urine	Finland	100	65	46	54	18%	27.7	24.3 ± 15.2 *				ng/mg	21	2
[88]	Control group	spot urine	Taiwan	27	49	0	27	55.6%	25.8	5 ± 4.92 *				µg/g	4	2
[89]	All population	spot urine	Japan	90	52	60	30	0%	25.2		5.8–23.2	0.90–48.0	9.3	ng/mg	9	3
[90]	Baseline value	spot urine	Canada	28	68.5	NA	NA	0%	27.1	10783 ± 5856 *				ng/g	9	2
[91]	control group baseline	spot urine	Spain	23	30.42	23	0	0%	25.32	9.29 ± 0.69 **				ng/mg	9	1
[91]	DHA group baseline	spot urine	Spain	23	29.97	23	0	0%	25.62	9.81 ± 0.79 **				ng/mg	9	1
[92]	Placebo group men	24 h urine	Canada	8	74.8	0	8	0%	25.9	8329 ± 3032 *				ng/g	8	1
[92]	Placebo group women	24 h urine	Canada	10	68.3	10	0	0%	25.2	11622 ± 4379 *				ng/g	7	1
[92]	Intervention group men baseline	24 h urine	Canada	11	71.8	0	11	0%	27.8	7245 ± 2703 *				ng/g	11	1
[92]	Intervention group women baseline	24 h urine	Canada	10	69.5	10	0	0%	25.5	7942 ± 3071 *				ng/g	7	1
[15]	Men group early morning	spot urine	Italy	22	34	0	22	38.1%	25.3		2.76–5.25		3.76	ng/mg	5	2

* SD; ** SEM.

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
