# Peer review of "Urinary 8-OHdG as a Biomarker for Oxidative Stress: A Systematic Literature Review and Meta-Analysis"

_ijms, 2020, doi:10.3390/ijms21113743_

Round 1

Reviewer 1 Report

In this manuscript, Hopf et al conducted systematic review of the literature and meta-analysis of the urinary levels of 8-oxo-deoxyguanosine (8-oxodG) in healthy individuals. Establishing the levels of 8-oxodG in general population has an importance for the future association studies aimed at revealing connection between oxidative stress and human disease. Additional need for this kind of analysis arises from the diversity of published data and techniques for the measurement of 8-oxoguanine.

The meta-analysis revealed that chromatography-based methods are more reliable than antibody assays. Also, authors provide several other important recommendations when and how to assess and report 8-oxodG levels. These recommendations will also aid others in reviewing scientific literature on urinary 8-oxodG.

There is one general question that was not addressed in the review. Since oxidative damage can be introduced during sample processing, could it contribute to the observed interquartile range and, also, differences between studies and, ELISA and chromatography based methods?

Recommended minor corrections:

  • Abstract, line 19: reactive oxygen species. Please, abbreviate as ROS.
  • Abstract, line 20: Correct to “DNA damage is”. The same for Introduction, line 39.
  • Introduction, line 41: metabolites
  • Introduction, line 60: has
  • Discussion, line 227: use “a comma” (,) after reference [97]

Reviewer 2 Report

This is a systematic review and meta-analysis to determine the range of 8-OHdG in urine from healthy people depending on its BMI or smoking habits. In general, the study does not provide changes in this parameter among the population studied. Probably, the inclusion of other studies with population suffering oxidative stress-associated diseases such as diabetes or metabolic syndrome could help to discern between normal levels range and levels associated with oxidative stress. 

Major points:

1.- The are not clear message of the study since no differences between levels of 8-OHdG have been found between population. As hypothesis, obese people could show higher levels of this marker in urine, however, in both cases, chemical and immunological determinations the range is lower in this population. For this reason, the inclussion of other population with oxidative-stress diseases would improve the study and determine if 8-OHdG in urine can be used as marker of oxidative stress.

2.- Authors recommend to measure 8-OHdG in spot morning urine, 12 and 24 hours urinary samples. However, they also indicate in the next sentence, that there is a good correlation between spot urinary samples and 24 h samples. Then, probably, the spot urinary sample determination must be the recommended sample for all the determinations to homongenize the studies as authors indicate in following sentences.

Minor points:

1.- Line 39, primarily instead of primerely.

2.-Lines 45-47, sentence is not clear. Please, clarify.

3.- Check line 559.

Round 2

Reviewer 1 Report

The authors have addressed my comments in full.

Reviewer 2 Report

I consider the study would be enriched with more studies, however, the message is clear and fine.